# QTL Mapping for Important Agronomic Traits Using a Wheat55K SNP Array-Based Genetic Map in Tetraploid Wheat

**DOI:** 10.3390/plants12040847

**Published:** 2023-02-14

**Authors:** Chao Ma, Le Liu, Tianxiang Liu, Yatao Jia, Qinqin Jiang, Haibo Bai, Sishuang Ma, Shuhua Li, Zhonghua Wang

**Affiliations:** 1State Key Laboratory of Crop Stress Biology for Arid Areas, College of Agronomy, Northwest A&F University, Yangling 712100, China; 2Agricultural Bio-Technology Research Center, Ningxia Academy of Agriculture and Forestry Science, Yinchuan 750002, China

**Keywords:** tetraploid wheat, 55K SNP array, agronomic traits, QTL cluster, marker segregation distortion, epistatic effect

## Abstract

Wheat yield is highly correlated with plant height, heading date, spike characteristics, and kernel traits. In this study, we used the wheat55K single nucleotide polymorphism array to genotype a recombinant inbred line population of 165 lines constructed by crossing two tetraploid wheat materials, Icaro and Y4. A genetic linkage map with a total length of 6244.51 cM was constructed, covering 14 chromosomes of tetraploid wheat. QTLs for 12 important agronomic traits, including plant height (PH), heading date (HD), awn color (AC), spike-branching (SB), and related traits of spike and kernel, were mapped in multiple environments, while combined QTL-by-environment interactions and epistatic effects were analyzed for each trait. A total of 52 major or stable QTLs were identified, among which may be some novel loci controlling PH, SB, and kernel length-width ratio (LWR), etc., with LOD values ranging from 2.51 to 54.49, thereby explaining 2.40–66.27% of the phenotypic variation. Based on the ‘China Spring’ and durum wheat reference genome annotations, candidate genes were predicted for four stable QTLs, *QPH.nwafu-2B.2* (165.67–166.99 cM), *QAC.nwafu-3A.1* (419.89–420.52 cM), *QAC.nwafu-4A.1* (424.31–447.4 cM), and *QLWR.nwafu-7A.1* (166.66–175.46 cM). Thirty-one QTL clusters and 44 segregation distortion regions were also detected, and 38 and 18 major or stable QTLs were included in these clusters and segregation distortion regions, respectively. These results provide QTLs with breeding application potential in tetraploid wheat that broadens the genetic basis of important agronomic traits such as PH, HD, AC, SB, etc., and benefits wheat breeding.

## 1. Introduction

Wheat is one of the most important food crops. With the global population expected to be more than 9 billion by 2050, the demand for wheat is also expected to increase by 60% [1,2]. Improving food yields remains an urgent task to achieve global food and nutrition security.

In recent years, researchers studying crop breeding have focused on the ongoing exploitation of novel genetic resources to increase crop yield potential. During the “Green Revolution” around the 1960s, the discovery and application of several genes related to crop dwarfism and dry matter accumulation increased the harvest index (HI), which not only led to significant increases in grain yield per unit area but also prevented global famine [3,4,5]. For more than half a century, the production of crops has been increasing. However, the widespread use of a small number of parents in modern breeding programs and the large-scale promotion of only a few superior varieties have resulted in the loss of wheat genetic resources, an increasingly narrow genetic basis, and the production growth into a bottleneck period [6,7,8]. The genome of hexaploid wheat is large (17.9 GB) and highly repetitive [9]; moreover, the genetic basis of quantitative traits such as yield and quality is complex and easily influenced by environmental factors. These factors have limited the cloning and application of genes/QTLs, making it difficult to rapidly develop genetic research and breeding for wheat, lagging behind maize and rice, which have relatively simple genomes [10,11]. Tetraploid wheat has two sets of seven homologous chromosomes (AABB, 2n = 4x = 28), a simpler genome, having numerous excellent germplasm resources and potentially superior genes [12,13]. The discovery and exploration of genetic resources for important agronomic traits in tetraploid wheat to improve genetic diversity will greatly benefit wheat breeding.

Wheat yield is strongly correlated with key agronomic traits, such as plant height, heading date, and spike and kernel characteristics. Plant height (PH) is a crucial agronomic feature in wheat, and an appropriate reduction in PH can reduce lodging and boost grain yield [14]. Twenty-five wheat semi-dwarfing genes have been cataloged, with multiple genes from tetraploid wheat, and the most widely used of these height-reducing genes are the *Rht-B1b* (*Rht1*), *Rht-D1b* (*Rht2*) and their orthologs, which are located on the chromosome of group 4 and encode variant DELLA proteins that result in reduced sensitivity to gibberellins (GA) [15,16].

The heading date (HD) is critical for wheat to adapt to harsh environments and maintain stable yields, not only determining the dispersion and geographical adaptation of the crop but also affecting yields [17]. More than a hundred QTLs for HD have been mapped across all wheat chromosomes [18]. Previous studies have shown that vernalization requirements and photoperiod sensitivity mainly influence and control the inheritance of HD [19]. *VRN1*, *VRN2*, *VRN3*, and *VRN4* are the major wheat vernalization (*VRN*) genes that affect the vernalization of wheat [20,21,22,23]. The photoperiod response is primarily regulated by three homologous dominant loci, *Ppd-A1*, *Ppd-B1*, and *Ppd-D1*, which are situated on chromosomes 2A, 2B, and 2D, respectively. The *Ppd1* gene functions as a regulator to reduce the delay of HD in wheat under short-day (SD) circumstances [24]. In addition, there are earliness *per se* (*Eps*) genes that have a strong impact on HD. The *Eps* genes were identified on chromosome 1$A^{m}$ in *Triticum monococcum*, which has significant epistatic interactions with photoperiod and vernalization, and significant interactions with environmental temperature [25].

The spike is one of the essential organs of wheat. Numerous reports have shown that spike morphological characteristics, including spike length (SL), spike-branching (SB), and kernel number per spike (KPS), are closely associated with yield [26,27,28]. *Q*, *compactum* (*C*), and *sphaerococcum* (*S1*) are three main genes related to the development of the common wheat spike, located on the long arm of chromosome 5A, chromosome 2D near the centromere, and chromosome 3D, respectively, which together play a part in the morphological development of spikes such as spike density and KPS, and also have pleiotropic effects on PH, grain shape, and seed threshability [29,30,31]. Tetraploid cultivars or varieties, however, do not have C or S genes because they lack D-genome chromosomes. Therefore, genes other than Q, C, or S or alleles of these three genes on homologous chromosomes could be responsible for the variance in the morphology of the tetraploid wheat spike [27,28,32].

The kernel traits include kernel length (KL), kernel width (KW), kernel area (Area), kernel perimeter (Peri), thousand kernel weight (TKW), etc. TKW, as one of the main components of wheat grain yield, has always been one of the critical objectives of breeding improvement, along with spikes per $m^{2}$ and kernels per spike. KL, KW, Area, etc. are highly relevant to TKW and directly affect the yield and end-use quality of wheat [33,34]. All 21 of wheat’s chromosomes have been found to have a number of QTLs associated with kernel characteristics in recent years [35,36,37]. However, many QTLs for TKW were too influenced by environmental factors to explain a majority portion of the phenotypic variation in kernel weight and also showed high QTL × genotype and QTL × QTL epistatic interactions [38,39]. Moreover, QTLs for other key kernel traits were rarely reported [33,40].

In this study, to gain more insight into the genetic basis of important agronomic traits in tetraploid wheat, 165 recombinant inbred lines (RILs) constructed from a cross between two tetraploid wheat materials were carefully selected as the mapping population. QTLs for 12 important agronomic traits, including plant height (PH), heading date (HD), spike length (SL), kernel number per spike (KPS), thousand-kernel weight (TKW), kernel length (KL), kernel width (KW), kernel length-width ratio (LWR), kernel perimeter (Peri), kernel area (Area), awn color (AC), and spike-branching (SB), were mapped using a novel genetic map constructed with the Wheat55K SNP array. Furthermore, combined QTL-by-environment interactions and epistatic effects of the major or stable QTLs identified were analyzed. The sequences of flanking SNP markers were blasted against the Chinese spring and durum wheat genome reference sequences, and the screened stable QTLs were further analyzed and predicted.

## 2. Results

### 2.1. Phenotypic Variation

Significant differences between parents were observed for several traits in multiple experimental environments (Table 1). Compared to Icaro, Y4 had higher PH, SL, KPS, TKW, KL, and KW, as well as longer lengths of each internode except I5L. On the contrary, Icaro had higher LWR and relatively shorter HD than those of Y4 (Table 1, Figure 1). Moreover, both parents also showed phenotypic variations in AC and SB. Icaro had black awns and normal morphology spikes, while Y4 had white awns and branched spikes (Figure 1b).

Seven traits (PH, HD, SL, TKW, LWR, AC, and SB) had relatively higher $H^{2}$ between 0.64 and 0.96, indicating that these traits were less influenced by environmental factors, while Area had the lowest $H^{2}$ at 0.20. The frequency distribution of HD and the six kernel-related traits (TKW, KL, KW, LWR, Peri, and Area) as well as the spike-related traits KPS and SL were approximately normally distributed in most experimental environments and the BLUE dataset, and transgressive segregations (the absolute values of both skewness and kurtosis are close to 1) in the direction of both parents were observed in the population, which were typical of quantitative trait characteristics (Appendix A). For PH, remarkable differences in height between population lines resulted in a bi-modal distribution, which allowed most lines to be classified as the dwarf and tall (Appendix A), suggesting that the height of the population may be primarily influenced by one major gene [41]. AC and SB could be divided into two opposing phenotypes that exhibit partially similar features to qualitative traits (Figure 1b, Appendix A).

### 2.2. Correlation Analyses among Different Traits

Based on BLUE values, the phenotypic correlations between the 12 agronomic traits are presented in Table 2.

As a result, PH was significantly and negatively correlated (*p* < 0.01) with HD, while it has nonsignificant correlations with both spike and kernel traits. SL was significantly and positively correlated with other spike traits KPS and AC, and negatively correlated (*p* < 0.01) with kernel traits KL, LWR, and Peri. KPS was significantly and negatively correlated with the kernel-related traits TKW, KL, KW, Peri, and Area, and significantly positively correlated (*p* < 0.01) with SB. Except for no significant correlation with Area, LWR had a significant negative correlation with KW and TKW and a significant positive correlation with KL and Peri (*p* < 0.01). The other kernel traits (TKW, KL, KW, Peri, and Area) were significantly positively correlated with each other (*p* < 0.01). The correlation coefficient between KL and Peri reached 0.974. The correlation coefficients between Area and KL, KW, and Peri ranged from 0.854 to 0.943.

### 2.3. Genetic Map Construction

All 165 lines of the RIL population were genotyped using the Wheat55K SNP array, and after the rejection of missing values and random removal of redundant markers using the BIN function of IciMappingV4.1, 1356 BIN markers were finally screened from 53,063 markers to construct the genetic map. This genetic map covered all 14 chromosomes of tetraploid wheat, spanning 6244.51 cM in length with an average marker density of 4.61 cM/SNP. The A genome contained 602 SNPs and the B genome had 754 SNPs. The total genetic distance of the A and B genome maps was 3259.14 cM and 2985.37 cM, respectively, and the marker densities were 5.41 cM/SNPs and 3.96 cM/SNPs, respectively (Appendix A).

### 2.4. Analysis of Segregation Distortions in the Genetic Map

It is well known that marker segregation distortion (SD) is a common phenomenon in genetic linkage maps construction and QTL mapping [42]. After excluding markers with severe distortion, a chi-square test was performed on the 1356 BIN markers involved in the construction of the genetic map, and if *p* < 0.05, the marker was considered to have undergone SD (significantly different from the expected 1:1 Mendelian separation ratio). The results show that a total of 837 (61.73%) BIN markers showed the distortion from Mendelian segregation. Among these distorted markers, 345 (57.31%) and 492 (65.25%) BIN markers showed SD in A and B genomes, respectively; the number of markers showing SD for each chromosome ranged from 19 (chr2A) to 90 (chr2B and 5B), and chromosomes 7A and 7B had the minimum (43.33%) and maximum (75%) marker distortion rate, respectively (Appendix A).

In the RIL population genetic map, regions consisting of five or more consecutive genetic markers with the same direction of segregation were generally referred to as segregation distortion regions (SDRs). The analysis of the SDR is also an essential metric for evaluating the genetic map. While 44 SDRs were detected on 13 chromosomes except for 2A, chromosome 4A had the largest number of SDRs (6 SDRs) (Appendix A). These SDRs comprised 470 SD markers with a total length of 2251.68 cM, accounting for 36.06% of the length of the genetic map, and most of them were biased towards the male parental (Y4) genotypes. Notably, the four adjacent SDRs (*AX-111495990*~*AX-110057517*, *AX-110935969*~*AX-110121036*, *AX-109346617*~*AX-110705049*, and *AX-111565010*~*AX-108891576*) on chromosome 4A deviate in the same direction, and this distribution form was also present on other chromosomes. The region on chromosome 7A from *AX-110526555* to *AX-110484953* consisted of 18 distorted markers biased towards the male parental genotypes and was the longest SDR with a length of 274.5 cM. The shortest SDR was located on chromosome 6A (*AX-108852957*~*AX-109515848*), which was 3.16 cM in length and consisted of six distorted markers (Appendix A).

### 2.5. QTL Mapping Analysis

The ICIM-BIP method was used to detect additive QTLs for the 12 target traits, and 152 additive QTLs were eventually detected (Appendix A). The LOD values of the QTLs ranged from 2.51 to 54.49 and were distributed on 14 chromosomes, explaining 0.45–66.27% of the phenotypic variation (Appendix A). Among the detected additive QTLs, the ones with LOD > 3 and explained phenotypic variation > 10% were defined as major QTLs. Additive QTLs that can be detected in two or more individual environments and BLUE dataset were also considered stable QTLs. There were 3, 3, 6, 4, 4, 7, 3, 5, 5, 3, 6, and 3 major or stable QTLs detected for HD, PH, SL, KPS, TKW, KL, KW, LWR, Peri, Area, AC, and SB, respectively (Table 3, Figure 2). These major or stable QTLs with LOD values ranging from 2.56 to 54.49 were distributed on 12 chromosomes and explained 2.40–66.27% of the phenotypic variation (Table 3).

Ten PH additive QTLs were mapped to chromosomes 2B (2), 3A (2), 4A (2), 6A (2), 6B (1), and 7A (1) (Appendix A). The LOD values of individual QTLs ranged from 3.04 to 19.53, and individual QTLs explained 3.78% to 23.02% of the PH phenotypic variation (Appendix A). *QPH.nwafu-6A.1* and *QPH.nwafu-6A.2* had LOD values > 3, explained > 10% of the PH phenotypic variation, and were considered the major QTLs. *QPH.nwafu-6A.1* and *QPH.nwafu-2B.2* were detected in at least 3 years and were considered stable QTLs (Table 3). Icaro contributed to the effect of a decreased PH for the two stable QTLs.

For HD, 10 additive QTLs were mapped to chromosomes 1B (1), 2A (2), 4A (1), 5A (2), 5B (2), and 6A (2) (Appendix A). The LOD values of individual QTLs ranged from 2.59 to 11.88, and individual QTLs explained 3.29% to 26.54% of the phenotypic variation (Appendix A). *QHD.nwafu-5A.1*, *QHD.nwafu-6A.1,* and *QHD.nwafu-6A.2* contributed more to the phenotype with LOD values ranging from 7.80 to 11.88, explaining 11.56% to 26.54% of the phenotypic variation (Table 3). The confidence interval of *QHD.nwafu-6A.1* was 198.5–201.5 cM (flanked by the SNP markers *AX-110419766* and *AX-111161134*) and overlapped with that of *QPH.nwafu-6A.1*. The genetic interval for *QHD.nwafu-6A.2* was between the markers *AX-111634639* and *AX-111800524*, coinciding with that of *QPH.nwafu-6A.2*. This suggests that the PH variation of these loci may be derived from the variation in HD.

Eleven additive QTLs for SL were mapped to chromosomes 2A (1), 3A (1), 4A (4), 4B (1), 5A (1), 5B (1), 6A (1), and 7A (1) (Appendix A). The LOD values of individual QTLs ranged from 3.04 to 19.53, and individual QTLs explained 3.78% to 23.02% of the phenotypic variation (Appendix A). Four major QTLs (*QSL.nwafu-2A*, *QSL.nwafu-3A*, *QSL.nwafu-4A.2*, and *QSL.nwafu-4A.3*) were detected with LOD values ranging from 3.04 to 19.53, explaining 10.25% to 23.02% of the phenotypic variation, of which *QSL.nwafu-3A* was detected in Y21, Y22, and BLUE dataset and was also considered as a stable major QTL. The allele for increasing SL in the locus was contributed by Y4 (Table 3).

Sixteen additive QTLs for KPS were mapped to chromosomes 1B (2), 2B (4), 3A (1), 4B (3), 5B (3), 6A (1), 7A (1), and 7B (1) (Appendix A). Individual QTLs explained 3.25% to 15.75% of the KPS phenotypic variation, with LOD values ranging from 2.53 to 10.32. *QKPS.nwafu-2B.3*, *QKPS.nwafu-4B.2*, *QKPS.nwafu-4B.3*, and *QKPS.nwafu-5B.3* contributed more to the phenotype with LOD values ranging from 4.16 to 10.32, explaining 5.51% to 15.75% of the phenotypic variation (Table 3). *QKPS.nwafu-2B.3* was mapped in Y21 and BLUE dataset, with an average LOD value of 7.24, and the favorable allele was contributed by Icaro, explaining an average of 8.08% of the phenotypic variation (Table 3).

The 19 additive QTLs of TKW were mapped to chromosomes 1A (1), 1B (3), 2A (2), 2B (3), 3A (1), 3B (2), 5A (1), 5B (3), and 6A (3) (Appendix A). The LOD values of individual QTL ranged from 2.57 to 15.18, and individual QTLs explained 2.97% to 17.05% of the phenotypic variation (Appendix A). Four major QTLs (*QTKW.nwafu-1B.1*, *QTKW.nwafu-2B.3*, *QTKW.nwafu-3B.2*, and *QTKW.nwafu-5B.2*) were detected, with LOD values ranging from 2.59 to 15.18, explaining 8.53% to 17.05% of the phenotypic variation, of which *QTKW.nwafu-1B.1* and *QTKW.nwafu-3B.2* were detected in both the one-year environment and BLUE dataset (Table 3).

Thirteen additive QTLs for KL were mapped to chromosomes 1A (1), 2B (2), 3A (1), 3B (1), 5B (3), 6A (1), 6B (1), and 7A (3) (Appendix A). The LOD values of individual QTLs ranged from 3.55 to 10.70, and individual QTLs explained 4.72% to 20.41% of the KL phenotypic variation (Appendix A). A total of seven major QTLs were detected, with LOD values ranging from 4.43 to 10.70, explaining 6.61% to 20.41% of the phenotypic variation (Table 3).

Eight additive QTLs for KW were mapped to chromosomes 1B (2), 3A (1), 4B (2), 6A (1), and 6B (2) (Appendix A). The LOD values of individual QTLs ranged from 2.55 to 6.82, and individual QTLs explained 5.14% to 11.44% of the phenotypic variation (Appendix A). *QKW.nwafu-3A*, *QKW.nwafu-6A,* and *QKW.nwafu-4B.2* contributed higher to the phenotypic variation, explaining 11.44%, 10.99%, and 11.11% of the KW phenotypic variation with LOD values of 3.91, 4.11 and 6.82, respectively (Table 3). Among the three major QTLs, two QTLs (*QKW.nwafu-3A* and *QKW.nwafu-4B.2*) had the favorable alleles contributed by Y4 and the remaining one QTL *QKW.nwafu-6A* had increasing KW alleles provided by Icaro.

Twenty-two additive QTLs for LWR were mapped to chromosomes 1B (4), 2B (2), 3A (2), 4A (1), 5A (1), 5B (6), 6A (1), 6B (1), and 7A (4) (Appendix A). The LOD values of individual QTLs ranged from 2.51 to 13.12, and individual QTLs explained 3.61% to 26.22% of the phenotypic variation (Appendix A). Three major QTLs were detected, with LOD values ranging from 2.56 to 13.12, explaining 8.96% to 26.22% of the phenotypic variation. In addition, *QLWR.nwafu-3A.2*, *QLWR.nwafu-7A.1*, *QLWR.nwafu-1B.4*, and *QLWR.nwafu-6B* were detected in multiple environments and considered stable QTLs (Table 3).

Nine additive QTLs for Peri were mapped to chromosomes 2B (1), 3B (1), 5B (3), 6A (2), and 7A (2) (Appendix A). The LOD values of individual QTLs ranged from 2.84 to 8.59, and individual QTLs explained 5.34% to 16.15% of the phenotypic variation (Appendix A). Five major QTLs were detected, explaining 12.57%~16.15% phenotypic variation (Table 3).

Ten additive QTLs for Area were mapped to chromosomes 1B (2), 2B (3), 3B (1), 5B (2), and 7B (2) (Appendix A). The LOD values of individual QTLs ranged from 2.53 to 8.45, and individual QTLs explained 2.97% to 15.40% of the phenotypic variation (Appendix A). *QArea.nwafu-1B.1*, *QArea.nwafu-1B.2,* and *QArea.nwafu-5B.2* contributed higher to the phenotype, with LOD values of 4.36, 7.33 and 8.45, explaining 12.90%, 13.51% and 15.40% of the phenotypic variation, respectively (Table 3).

Eighteen additive QTLs for AC were mapped to chromosomes 1A (1), 1B (4), 2B (2), 3A (2), 3B (1), 4A (2), 4B (1), 5A (1), 6A (2), and 7A (2) (Appendix A). The LOD values of individual QTLs ranged from 2.57 to 54.49, and individual QTLs explained 0.45% to 31.77% of the phenotypic variation (Appendix A). Three major QTLs were detected with LOD values ranging from 4.56 to 54.49, explaining 5.38% to 31.77% of the phenotypic variation. *QAC.nwafu-3A.1*, *QAC.nwafu-3A.2*, *QAC.nwafu-4A.1*, *QAC.nwafu-1B.3*, and *QAC.nwafu-1B.4* were detected in 2 years and were therefore considered to be stable QTLs (Table 3).

Six additive QTLs for SB were mapped to chromosomes 2B (2), 4A (1), 4B (1), 5B (1), and 7A (1) (Appendix A). The LOD values of individual QTLs ranged from 2.63 to 53.74, and individual QTLs explained 1.33% to 23.16% of the phenotypic variation (Appendix A). *QSB.nwafu-2B.1*, *QSB.nwafu-2B.2*, and *QSB.nwafu-5B* were the major QTLs with LOD values ranging from 3.33 to 53.74, explaining 8.97% to 23.16% of the phenotypic variation. In addition, *QSB.nwafu-2B.1* and *QSB.nwafu-2B.2* were detected in Y21 and Y22, which were also considered stable QTLs (Table 3).

### 2.6. QTL Cluster Analysis

In most crops, some QTLs associated with agronomic traits are distributed in clusters on chromosomes, manifesting as QTLs that control different traits located within the same marker interval or adjacent regions [11,43,44]. This distribution pattern was also observed in the tetraploid wheat RIL population via analysis of QTL mapping results in this study. A total of 31 QTL clusters were mapped to 12 chromosomes except for 1A and 5A (Appendix A). These QTL clusters contained 97 QTLs, of which 38 were either major or stable QTLs, accounting for about 73% of the number of major and stable QTLs detected using the ICIM-BIP method. Twenty QTL clusters contained major or stable QTLs. The QTL cluster at 260–268 cM on chromosome 2B (C21) contained three QTLs (*QTKW.nwafu-2B.2*, QKPS.nwafu-2B.4, and QArea.nwafu-2B.3) associated with TKW, KPS, and Area, and three stable QTL *QSB.nwafu-2B.1*, *QKPS.nwafu-2B.3* and *QSB.nwafu-2B.2* were also located in this cluster (Appendix A). Two stable QTL *QSL.nwafu-6A* and *QPH.nwafu-6A.1* and one major QTL *QHD.nwafu-6A.1* were genetically located close to each other and were also detected to be in the same cluster (C8) (Appendix A). The QTL clusters at 260–268 cM on chromosome 2B (C21) and 118-143 cM on chromosome 5B (C26) both contained the largest number of six QTLs controlling four traits including SB, KPS, TKW, and Area and five traits including Area, SB, LWR, TKW, and KPS, respectively (Appendix A).

According to the findings, various yield traits such as KPS, SL, SB, and kernel-related traits were significantly correlated with each other (Table 2). Moreover, the positions of QTL for these traits tended to be close to each other (Appendix A). Most QTL clusters (C2, C3, C6, C7, C10–17, C19–28, C30, and C31) also tended to contain multiple QTLs for yield traits (Appendix A), which may be caused by the presence of pleiotropism at some loci, where a single gene directly or indirectly affects multiple traits in the spike and/or the kernel. The significant negative correlation between PH and HD in the correlation analysis based on the BLUE dataset (*p* < 0.01, Table 2) also led to the speculation that the genes or QTLs controlling PH and HD might be tightly linked in position or one gene affecting both PH and HD traits, thus showing a close genetic correlation; this speculation was further verified by the tightly linked distribution of QTLs for PH and HD on C5, C8, and C9 (Appendix A). Moreover, we have found the same phenomenon in many previous reports [45,46,47].

### 2.7. Combined QTL–Environment Interaction Analysis

The MET-Add method was used for cQTL analysis, and a total of 184 additive cQTLs for PH (23), HD (13), SL (8), KPS (12), TKW (15), KL (23), KW (10), LWR (27), Peri (13), Area (12), AC (22), and SB (6) were mapped (Appendix A). Most of these cQTLs were minor QTLs. Moreover, 41 of the 52 major or stable QTLs detected using the ICIM-BIP method were similarly detected using the MET-Add method (Table 3 and Table 4), showing that they might be stable in expression and less susceptible to environmental influences.

Among the cQTLs for PH, PVE (A) (phenotypic variation explained by additive and dominant effects) was 2.61–51.32%, while PVE (A by E) (additive and dominance by environment effects for corresponding QTLs) was 0.28–2.25% (Appendix A), with a significant decrease in PVE (A by E) compared to PVE (A), demonstrating that PH was primarily influenced by genetic factors. This also coincided with the estimated H^2 of 0.96 for PH based on the BLUE dataset (Table 1). In contrast, except for AC and SB, the other nine traits were relatively susceptible to environmental factors.

### 2.8. Epistatic Analysis

A total of 539 digenic epistatic QTLs (eQTLs) were detected on 14 chromosomes, explaining 0.31% to 7.31% of phenotypic variation (Figure 3, Appendix A). For PH, HD, SL, KPS, TKW, KL, KW, LWR, Peri, Area, AC, and SB, 96, 21, 3, 41, 34, 41, 39, 60, 38, 64, 98, and 4 digenic epistatic QTLs were detected, respectively (Appendix A). Of these, 51.8% of eQTLs had negative epistatic effects (Appendix A), with negative values indicating that the recombinant genotype outperformed the parental genotype in terms of epistatic impact.

The epistasis of these six traits (PH, SL, KL, KW, LWR, and AC) were characterized by additive interactions between additive QTLs and random loci and between multiple random loci. Some of the major or stable QTLs detected using the ICIM-BIP method also showed epistatic effects (Appendix A). These results showed that these traits were influenced by both epistatic and additive genetic effects. However, for HD, KPS, TKW, Peri, Area, and SB, numerous eQTLs were not detected in the BIP and MET-Add analyses, suggesting that these random loci may not have a direct effect on the phenotype but can indirectly influence it through interactions between loci.

### 2.9. Effects of Stable Quantitative Trait Loci on Related Traits

There were 184 cQTLs for the 12 traits identified by combined QTL and environment interactive effect analysis (Appendix A). Four of them were identical to the four stable QTLs (*QPH.nwafu-2B.2*, *QAC.nwafu-3A.1*, *QAC.nwafu-4A.1*, and *QLWR.nwafu-7A.1*), which were identified via single-environment analysis on chromosomes 2B, 3A, 4A, and 7A, and had relatively high PVE and narrow genetic localization intervals (Figure 2, Table 3 and Table 4). To assess the effects of different haplotypes on the traits of interest, lines with the corresponding homozygous alleles in the population were obtained based on the genotyping results of the two flanking markers of these QTLs. Depending on whether the homozygous alleles of the QTL locus were from Icaro or Y4, the RILs were assigned into two groups accordingly: Icaro type and Y4 type. In addition, the corresponding phenotypes of the two groups were analyzed.

For PH, one stable QTL *QPH.nwafu-2B.2* and one stable major QTL *QPH.nwafu-6A.1* were identified. Moreover, significant differences in PH were observed between the two groups of lines corresponding to each locus (*p* < 0.01, Figure 4). The PH of Icaro type lines was significantly lower than those Y4 type lines, and the effect of the *QPH.nwafu-6A.1* was greater than that of *QPH.nwafu-2B.2* in reducing the magnitude of PH. To distinguish more carefully the effects of the two loci on PH as well as HD and TKW, the lines were further divided into four groups based on the different genotype combinations of the two loci (Figure 5). Compared with lines carrying Y4 alleles at both *QPH.nwafu-2B.2* and *QPH.nwafu-6A.1*, each locus carrying Icaro alleles was able to reduce PH by 3.97% to 9.76% and 18.98% to 24.92%, respectively, and clustering of the two further contributed to this effect, reaching 26.53% to 28.10% (Figure 5a). For HD, the two loci show different effects. For the *QPH.nwafu-6A.1*, the HD of lines carrying Icaro alleles were significantly delayed in all environments and BLUE dataset (Figure 5b). However, for the *QPH.nwafu-2B.2*, the HD of lines carrying Icaro alleles was slightly earlier; in addition, neither locus had a significant positive or negative effect on TKW (Figure 5c).

As a stable locus, *QLWR.nwafu-7A.1* was identified under the Y21, Y22, and BLUE datasets (Table 3). For the *QLWR.nwafu-7A.1*, the two groups of RILs also showed significant differences in LWR (*p* < 0.01, Appendix A). The LWR of Y4 type lines had a significantly larger LWR compared with those Icaro type lines; however, the TKW was not significantly affected. For the *QAC.nwafu-3A.1* and *QAC.nwafu-4A.1*, lines carrying from Icaro alleles had a higher penetrance of black AC, and clustering of the two loci would further enhance the penetrance (Appendix A).

Spike-branching phenotype on tetraploid wheat is an important additional spikelet trait associated with a significant increase in KPS; thus, we also performed a preliminary effect analysis on the two identified SB stable loci. For the *QSB.nwafu-2B.1* and *QSB.nwafu-2B.2*, the lines were also further classified into four groups based on the different genotype combinations of the two loci. Compared with lines carrying Y4 alleles at both *QSB.nwafu-2B.1* and *QSB.nwafu-2B.2*, each locus carrying Icaro alleles was able to increase KPS, and clustering of the two could further contribute to this effect (Appendix A); however, the increase did not all reach statistical significance in some of the environments. In addition, TKW was not significantly influenced by different genotypes (Appendix A), indicating that these two QTLs could increase KPS without affecting kernel weight.

### 2.10. Genetic Analysis of QPH.nwafu-2B.2, QAC.nwafu-3A.1, QAC.nwafu-4A.1, and QLWR.nwafu-7A.1

*QPH.nwafu-2B.2* was mapped in the region from 165.67 to 166.99 cM on chromosome 2B, corresponding to the physical region of 663.7 to 664.2 MB in the Chinese Spring (CS) v1.0 genome and 652.8 to 653.3 MB in the Durum Wheat (cv. Svevo) genome. A total of 15 and 14 genes were annotated in these two genomes (Figure 6a). *QAC.nwafu-3A.1* was mapped in the region from 419.89 to 420.52 cM on chromosome 3A, corresponding to the physical region of 22.2 to 22.3 MB and 18.8 to 18.9 MB in the CS and Durum Wheat genome, respectively (Figure 6b). *QAC.nwafu-4A.1* was mapped in the region from 424.31 to 447.4 cM on chromosome 4A, corresponding to the physical region of 2.9 to 4.2 MB and 0.04 to 1.18 MB in the CS and Durum Wheat genome, respectively (Figure 6c). *QLWR.nwafu-7A.1* was mapped in the region from 166.66 to 175.46 cM on chromosome 7A, corresponding to the physical region of 652.3 to 661.2 MB and 646.5 to 654.6 MB in the CS and Durum Wheat genome, respectively (Figure 6d). Based on the reference genomes of CS, a total of 3, 44, and 183 genes were annotated in the corresponding physical region of *QAC.nwafu-3A.1*, *QAC.nwafu-4A.1*, and *QLWR.nwafu-7A.1*, respectively. In addition, based on the reference genomes of Durum Wheat, a total of 8, 84, and 350 genes were annotated in the corresponding physical region of *QAC.nwafu-3A.1*, *QAC.nwafu-4A.1*, and *QLWR.nwafu-7A.1*, respectively.

## 3. Discussion

### 3.1. Analysis of the Relation of Mapped QTLs to Those Found in Previous Studies

We identified loci for some of the major agronomic traits, both those already reported and confirmed, and novel loci with promising applications. These loci can subsequently develop specific molecular markers capable of efficiently targeting genes for marker-assisted selection in wheat breeding. It should be noted that most studies to date have focused on common wheat, whereas tetraploid wheat may exhibit different compositions for these complex traits. Thus, we focused mainly on the previous reports on tetraploid wheat compared with the QTLs in Table 3 (Table 5).

Plant height is a quantitative trait controlled by multiple genes/QTLs. In the present study, we identified two stable QTLs, *QPH.nwafu-6A.1* and *QPH.nwafu-2B.2*, that simultaneously affect HD and PH (Figure 5). *QPH.nwafu-6A.1* and *QHD.nwafu-6A.1* were mapped to the same interval (197.82–201.86 cM on 6A), which overlapped with *QSL.nwafu-6A* (196.86–201.86 cM on 6A), indicating that these three traits might be contributed by a single gene/locus on 6A chromosome. However, the semi-dwarf durum cultivar Icaro has previously been reported by Tang [48] to contain *Rht18* (412–445 Mb) within the physical interval corresponding to the *QPH.nwafu-6A.1*. Furthermore, Icaro type lines at the *QPH.nwafu-6A.1* locus were approximately 27% shorter in height compared to the tall progeny (Figure 5), which is essentially the same as the result reported by Tang et al. [49] that *Rht18* reduced PH by 26%. These results all seem to indicate that *QPH.nwafu-6A.1* is the *Rht18* that has been mapped and has a significant impact on the PH phenotype. *Rht18* was localized in the region from 412 to 445 Mb on chromosome 6A, which contains the *GA2oxA9* encoding a GA 2-oxidase, and overexpression of *GA2oxA9* reduced bioactive GA content and PH [50,51]. Both *QPH.nwafu-2B.2* and another semi-dwarfing gene *Rht4*, which is tightly linked to the microsatellite marker WMC317, are located on 2BL. However, BLAST analyses revealed that the positions of the two loci are not identical (Figure 6a). In addition, Icaro type alleles at the *QPH.nwafu-2B.2* locus had a tendency to accelerate HD and reduce PH, which is similar to the function of the photoperiod response locus *Ppd-D1a* [52]. However, it also differs from the position of the homologous locus of *Ppd-D1a* on chromosome 2B (*Ppd-B1*) [53]. Therefore, *QPH.nwafu-2B.2* may be a new locus for PH. Both the shortened heading date and the reduced plant height facilitate the transport and distribution of assimilates produced by the plant to the seeds, spikes and other “sink”, and the increase in assimilates admitted to the “sink” can help improve crop yields under a certain number of “sources”; therefore, *QPH.nwafu-2B.2* is a highly promising locus for application, and further research is necessary.

*QHD.nwafu-5A.1* shared the same locations as previously reported QTLs [45,46,54], and explained 14.88% of the HD variation (Table 3 and Table 5). Analysis of the physical location of the flanking markers indicated that the region contained the two reported vernalization genes *TaVRN1-5A* and *TaVRN2-5A* [20,55] and a flowering gene *WSOC1-5A* [56], so they are likely candidates.

The stable major QTL *QSL.nwafu-3A* detected under the Y21, Y22, and BLUE dataset shared the same position as the locus reported by Giraldo et al. [57]. *QKPS.nwafu-4B.3* was consistent with the locus reported by Kidane et al. [58]. *QKPS.nwafu-2B.3* and *QKPS.nwafu-5B.3* had extensive overlap with the two loci reported by Mangini et al. [39], respectively. The TKW-related physical intervals reported by Peng et al. [54], Graziani et al. [59], and Roncallo et al. [45] contained *QTKW.nwafu-1B.1*. The physical interval of *QTKW.nwafu-2B.3* contained multiple previously reported TKW loci [39,60,61]. The physical location of *QTKW.nwafu-5B.2* had a large overlap with the locus of TKW reported by Roncallo et al. [45] and Peng et al. [54]. They may be influenced or controlled by the same gene. Ma et al. [62] identified *TaGS5* on 3B, but this gene’s location was different from that of *QTKW.nwafu-3B.2*, suggesting that there may be novel genes associated with TKW in the QTL. In addition, compared with the physical map of QTLs for other yield-related traits such as KL, KW, LWR, and kernel size (KS) [37,63], the physical regions of QTLs in this study did not overlap with these loci.

**Table 5 plants-12-00847-t005:** QTL mapped in this study compared with those previously reported.

				Near Locus in Previous Studies		
Traits	QTL	PVE (%)	Physical Position (MB)	Marker Interval	Physical Position (MB)	Reference
**PH**	QPH.nwafu-6A.1	51.10(Y19)/61.76(Y20)/66.27(Y21) /38.24(Y22)/63.87(BLUE)	97.83–454.64	IWA3230-IWB62878 (*Rht18*)	412–445	[48]
**HD**	QHD.nwafu-5A.1	14.88(Y21)/16.49(BLUE)	493.27–661.33	IWB4912-IWB38386	465.95–535.43	[54]
				IWA7665-IWB59530	512.12–527.66	[46]
				IWB42031-IWB38885	587.19–612.37	[46]
				IWA7162-wmc727	590.23–661.71	[45]
				IWB51362-IWB68625	625.60–640.51	[46]
				IWB14680-IWB11691	639.17–653.03	[46]
				wmc727-IWB28907	645.16–661.37	[46]
**SL**	QSL.nwafu-2A	14.13(Y22)/10.25(BLUE)	319.88–532.72	IWB51736-IWB66316	522.21–605.33	WheatOmics 1.0
	QSL.nwafu-3A	23.02(Y21)/12.06(Y22)/16.93(BLUE)	714.71–736.11	IWB12017-IWB67819	727.73–735.81	[57]
	QSL.nwafu-4A.2	20.47(Y21)	621.84–626.80	IWB57472-IWB71581	625.84–641.73	WheatOmics 1.0
	QSL.nwafu-4A.4	5.76(Y22)/4.94(BLUE)	530.42–536.90	IWA7521-IWA2585	59.48–536.90	WheatOmics 1.0
**KPS**	QKPS.nwafu-2B.3	10.64(Y21)/5.51(BLUE)	100.91–128.12	IWB26450-IWB44316	99.22–172.75	[39]
	QKPS.nwafu-4B.3	12.09(BLUE)	487.64–658.68	IWB53931-IWB73383	616.76–632.99	[58]
				IWA1861-wPt-5265	651.91–668.84	[39]
				IWB72179-IWB4448	651.91–658.51	WheatOmics 1.0
	QKPS.nwafu-5B.3	15.75(Y20)	490.91–504.27	IWB56759-IWB35093	475.37–510.43	[39]
**TKW**	QTKW.nwafu-1B.1	17.05(Y22)/8.53(BLUE)	314.29–319.37	gwm264-GWM274	33.99–594.22	[54]
				IWB9645-IWA3502	53.20–464.24	[59]
				IWB9083-IWB6898	88.74–396.68	[45]
	QTKW.nwafu-2B.3	11.45(BLUE)	30.17–772.62	IWA7916-IWB44381	53.45–63.97	R. Patil et al. Euphytica 190 117–129 (2013)
				IWB26450-IWB44316	99.22–172.75	[39]
				IWA5436-IWA7019	196.55–537.61	[60]
				IWB49384-IWB7671	696.91–749.58	[61]
				wPt-3651-IWB29332	702.45–733.56	Z. Peleg et al. J. Exp. Bot. 62 5051–5061 (2011)
				IWA3474-IWB55786	765.31–789.41	[45]
	QTKW.nwafu-5B.2	11.11(Y21)	512.72–529.97	IWB66909-wPt-3030	509.55–537.85	[45]
				IWB72334-IWB65694	512.00–537.85	[54]

Spike-branching (SB) phenotype can alter the capacity of the “sink” by changing spike traits (e.g., spikelet number per spike and KPS), which has the potential to increase grain yield. In this study, two stable loci of SB (*QSB.nwafu-2B.1* and *QSB.nwafu-2B.2*) located on chromosome 2B were identified, explaining 10.00–11.20% and 8.97–23.16% of the phenotypic variation, respectively. Both loci had favorable alleles for SB traits contributed by Y4 (Table 3). Previous studies have shown that an amino acid substitution in the AP2/ERF domain of *TtBH-A1*, representing a mutant allele at the *WFZP-A* locus, causes spike branching in tetraploid *T. turgidum* wheat [64]. The expression of the trait is influenced by various environmental factors, in addition to the fact that *wfzp-A/TtBH-A1* is the main genetic factor determining the spike branching of *turgidum* type in tetraploid wheat, and other genetic factors also play a role in regulating *turgidum*-type SB [65,66,67]. Wolde et al. [26] reported the new modifier QTL *QSS.ipk-2BS* for spike-branching on 2BS, which was highly associated with coding sequence variation of the homologous allele of *TtBH-B1* (*b*$h^{t}$*-B1*). However, BLAST analysis showed that *QSB.nwafu-2B.1* and *QSB.nwafu-2B.2* were located on the long arm of chromosome 2B, which was far from *QSS.ipk-2BS*. This suggests that these two QTLs are potentially new SB loci.

### 3.2. Segregation Distortion Regions in the Genetic Map

It has generally been assumed that many factors from gametogenesis to zygote formation, and then to the post-zygote stage can lead to segregation distortion [68]. Marker segregation distortion is a common biological phenomenon [69]. However, these markers are either simply discarded or incorporated unnoticed in most studies and applications [70]. Hackett and Broadfoot [71] concluded from simulation analysis that the presence of segregation distortion had little effect on the accuracy of marker order or genetic map length. Zuo et al. [70], based on a study of 519 soybean RIL populations from orthogonal and reciprocal crosses between LSZZH and NN493-1, found that the inclusion of distorted markers had a positive effect on increasing genome coverage and improving the concordance of linkage maps with genome. Therefore, based on the results of previous studies, we believed that when the exact chromosome to which an abnormal marker belongs is known and anchored during the creation of a genetic map, the appropriate inclusion of slightly distorted markers not only reduces the effect of marker positional disorder but also increases the marker density of the genetic map.

Forty-four SDRs were detected with a length of 2251.68 cM, accounting for 36.06% of the overall length of the genetic map, distributed over all chromosomes except 2A (Appendix A). Eighteen major or stable QTLs were located in the SDRs, accounting for 34.62% of the total number of major and stable QTLs detected. A few major or stable QTLs for all traits except AC, SB, and KW were also distributed in SDRs (Table 3, Appendix A). Of the 837 distorted markers, 555 (66.3%) and 282 (33.7%) were biased toward the male and female parental genotypes, respectively. In addition, 31 of the 44 SDRs were also biased toward the male parental genotype (Appendix A). The distribution and biased direction of distorted markers were more inclined to the B genome and male parental genotypes, respectively. The occurrence of segregation distortion in hybrid progenies makes the allele frequencies from both parents change in the progeny population. The changing allele frequencies indicate the evolution of the population, which has important evolutionary significance [72,73]. Thus, changes in allele frequencies due to selection and other factors, the impact of such changes on estimating genetic linkage distances between loci, and how to efficiently select parental material and loci based on the direction of distortion bias should also be critical issues to consider in future breeding practices.

### 3.3. Candidate Gene Analysis of QPH.nwafu-2B.2, QAC.nwafu-3A.1, QAC.nwafu-4A.1, and QLWR.nwafu-7A.1

Fifteen genes were annotated in the chromosome 2B region between 663.7 and 664.2 MB, some of which may be associated with PH (Figure 6a, Appendix A); for example, *TraesCS2B01G468100* encodes the WD40 protein, which is involved in several cellular processes including signal transduction, transcriptional modulation, and histone modification, and was considered a key regulator of plant developmental processes [74]. Moreover, five genes, such as *TraesCS2B01G467800*, *TraesCS2B01G467700,* and its homologous gene *TRITD2Bv1G217740*, all encode MYB transcription factors that play significant roles in plant developmental processes and stress responses. AtMYB68, a root growth-specific regulator in *Arabidopsis thaliana*, influences overall plant development in unfavorable situations (such as elevated temperatures) [75]; during the seedling stage, AtMYB38 and AtMYB18/LAF1 regulate the hypocotyl response to blue and far-red light, respectively [76,77].

*TRITD4Av1G000740*, located within the 0.04–1.18 MB candidate region of the AC trait (Figure 6c, Appendix A), encodes a basic helix loop helix (bHLH) DNA-binding superfamily protein G; *TraesCS4A01G006100* and its homologous *TRITD4Av1G000100* encode an MYB transcription factor (TF). Notably, *bHLH* is a regulatory gene for anthocyanin biosynthesis, and MYB-TFs also play an essential role in the regulatory network for its biosynthesis [78]. In monocotyledons such as *Zea mays* and *Oryza sativa*, MYB-TFs co-regulate anthocyanin biosynthetic enzymes (e.g., *CHS*, *F3H* and *DFR*, *LDOX*, *BAN/ANR*, *UFGT*) with other TFs [79,80].

A total of 183 genes were annotated in the 652.3–661.2 MB region of 7A, several of which may be associated with LWR traits (Figure 6d, Appendix A). For example, *TraesCS7A01G457500* and the homologous *TRITD7Av1G244030* encode an F-box protein that is involved in many pathways of plant nutrition and reproduction [37,81]. *TraesCS7A01G461700* and its homologous *TRITD7Av1G245930* encode an Auxin response factor. ARF family TFs play a key role in transmitting auxin signals to alter plant growth and development [82]. In rice, increased auxin content induces auxin response factor-mediated activation of N$O_{3}$^−^ transporter and N-metabolism genes, resulting in improved N-use efficiency and grain yield [83].

## 4. Materials and Methods

### 4.1. Plant Materials and Field Trials

The mapping population was from the 165 $F_{6}$ generation RILs derived from a cross between the durum wheat variety named Icaro and a variety of *T. turgidum* ssp. *turgidum* known as Y4 that was selected and bred in our laboratory. Icaro has an earlier maturity, black awn, and disease resistance, and Y4 has larger and fuller seeds, higher PH, longer spike with spike-branching characteristics, white awn, and the stalks are so tough that they do not fall over easily.

RILs and their parents were grown in one row for each line at Yangling (108°07′ E, 34°30′ N) during the 2018–2019 (Y19), 2019–2020 (Y20), 2020–2021 (Y21), and 2021–2022 (Y22) growing seasons. Before sowing, tilling and basal fertilizer were applied to the experimental field; conventional weeding and winter irrigation were carried out during plant growth. Three replications were used in the randomized block design of the experiment. Each line was planted with 15 seeds evenly spaced along a 1.5 m row, with 0.25 m between each row.

### 4.2. Phenotypic Evaluation and Statistical Analysis

The following 12 important agronomic traits were investigated at different growth stages. Among them, HD was investigated at the early heading stage, and PH, SB, and AC were investigated at the maturing stage; the rest were measured after harvest.

HD was recorded when half of the plant spikes of each line emerged from the flag leaf sheath and converted to a representation of the number of days difference between it and the earliest HD in the population. Five representative plants of each line were randomly selected for manual measurements of PH, SL, and KPS. PH was measured from the ground to the top of the main spike (excluding awns). For SL, the length from the base of the rachis to the tip of the terminal spikelet was measured, excluding the awns. KPS was obtained from five spikes that had previously been measured for SL, individually threshed per spike and counted for kernel number. The other kernel-related traits, including TKW, KL, KW, LWR, Peri, and Area, were measured in the laboratory using an SC-G automatic grain analyzer (Wanshen Testing Technology Co., Ltd., Hangzhou, China). The AC and SB phenotypes for each line were typed according to a rank of 0 and 2, where 0 was given to a spike-branching or black-awn phenotype and 2 to a non-branching or white-awn phenotype.

The PH and AC data were collected from Y19 to Y22, and the HD, SL, KPS, TKW, KL, KW, LWR, Peri, Area, and SB data were collected from Y20 to Y22. For each trait, best linear unbiased estimation (BLUE) was calculated across environments using R package “lme4” and used for the estimation of genotype fixed effects and the calculation of broad-sense heritability ($H^{2}$). Based on the BLUE dataset, descriptive statistics, analysis of variance (ANOVA), and Pearson correlation analysis were performed for the target traits using SPSS 23.0 (https://www.spss.com, accessed on 2 September 2022). Population phenotype frequency distributions and the boxplot were drawn using Origin 2021 (https://www.originlab.com, accessed on 20 October 2022).

### 4.3. Genotyping and Genetic Map Construction

The genomic DNA (gDNA) of each RIL and parent lines was extracted from fresh leaves following the modified cetyl trimethylammonium bromide (CTAB) extraction method [84]. After DNA integrity, concentration and purity were confirmed, and the gDNA was hybridized to the Wheat55K SNP array containing 53,063 SNP markers. Genotyping was performed by Beijing CapitalBio Technology Co., Ltd. (https://www.capitalbiotech.com/, accessed on 7 September 2022).

The polymorphic SNPs with heterozygous genotypes between parents or allele frequencies <0.3 or >0.7 among the RILs were removed. The IciMappingV4.1 (https://isbreeding.caas.cn/, accessed on 10 September 2022) BIN module was used to remove redundant SNPs and SNPs with missing rate >10%, distortion *p* value >0.01, and obtained BIN markers. The BIN markers were then used for map construction. Briefly, the BIN markers were firstly anchored to the corresponding chromosomes by combining data on the corresponding physical positions of the markers in the Chinese Spring (CS) reference genome (IWGSC RefSeq v1.0, https://urgi.versailles.inra.fr/blast_iwgsc/blast.php, accessed on 2 October 2022). Ordering and Rippling of markers were then performed using the nnTwoOpt algorithm and the SARF criterion, respectively. Recombination frequencies were converted into genetic distance (cM, centimorgan) using the Kosambi mapping function. The genetic linkage map was visualized using Mapchart V2.3.

### 4.4. Quantitative Trait Loci Mapping

QTL analysis was undertaken for the 12 agronomic traits using the inclusive composite interval mapping (ICIM) method in the biparental population BIP module of IciMappingV4.1 with the following parameters: Step = 1 cM, PIN = 0.0001, and the LOD threshold for a significant QTL was set to 2.5 [85,86]. For each trait, QTLs less than 1 cM apart or sharing a common flanking marker were regarded as a single QTL. The multi-environment trials (MET-Add) module of IcimappingV4.1 was used to identify cQTL (combined quantitative trait loci) and additive-by-environment (A by E) interactive effects [87]. Epistatic analysis was performed using the IciMappingV4.1 MET-EPI module, and eQTL (epistatic quantitative trait loci) detection parameters were set as follows: LOD = 5, step = 5 cM, PIN = 0.0001. QTLs were named based on the International Rules of Genetic Nomenclature, where “nwafu” represents Northwest Agriculture and Forestry University.

Flanking molecular marker sequences of the major or stable QTLs were compared with the CS reference genome (IWGSC RefSeq v1.0) and the durum wheat genome reference sequence (Durum Wheat Svevo Refseq Rel. 1.0) on the Triticeae Multiomics Center website (http://202.194.139.32/, accessed on 10 October 2022) to obtain the corresponding physical interval locations. Moreover, Uniprot was used (https://www.uniprot.org/, accessed on 10 October 2022) for further annotation and functional analysis of the candidate genes in the QTL intervals.

## Figures and Tables

**Figure 1 plants-12-00847-f001:**
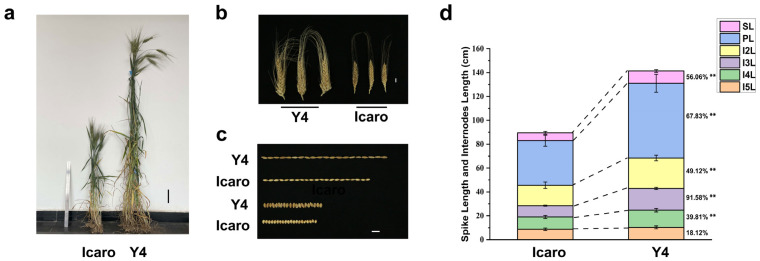
Phenotypic characteristics of the parent ‘Icaro’ and ‘Y4′. (**a**) Plant height phenotype. The black line represents the scale = 10 cm. (**b**) Spike characteristics. Scale bar = 1 cm. (**c**) Kernel phenotypes. Scale bar = 1 cm. (**d**) Spike length and internodes length. SL, spike length; PL, peduncle length; I2L, I3L, I4L, and I5L represent the length of the second, third, fourth, and fifth internode from the top, respectively. ** significant at the level of *p* < 0.01.

**Figure 2 plants-12-00847-f002:**
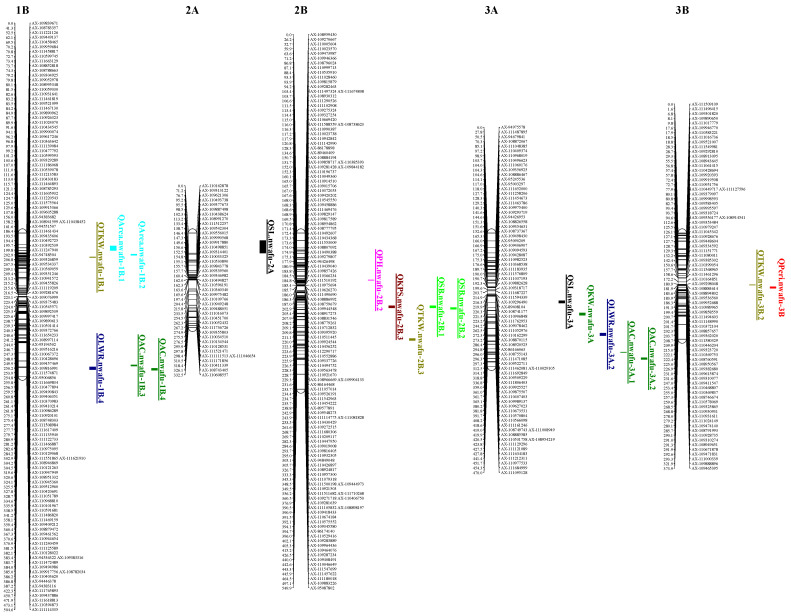
Chromosome positions of the major or stable additive QTLs for the 12 traits described in Table 1. The underlined QTLs indicate that they are the stable QTLs identified in this study. Different colors are used to distinguish different traits on the same chromosome.

**Figure 3 plants-12-00847-f003:**
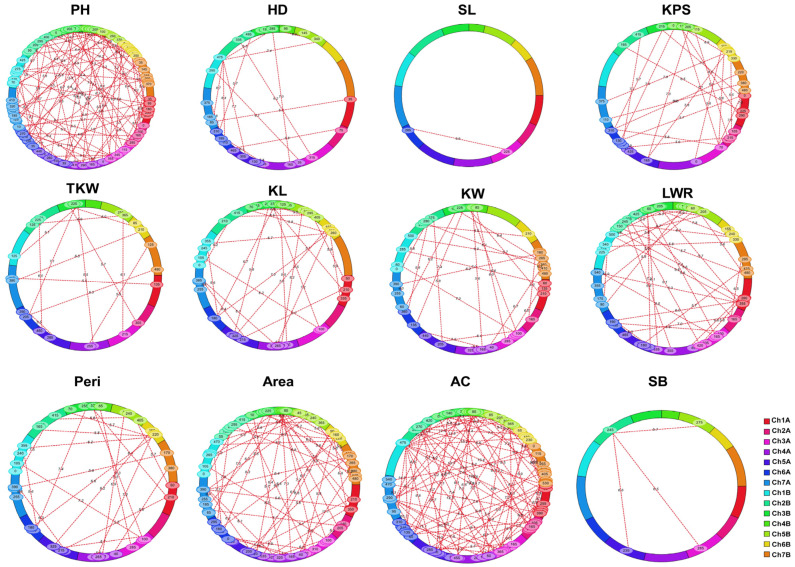
Detection of epistatic QTLs for the 12 traits. LOD value = 5.5.

**Figure 4 plants-12-00847-f004:**
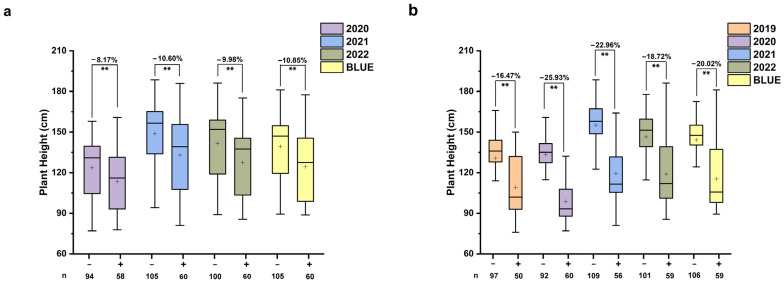
Effect of *QPH.nwafu-2B.2* (**a**) and *QPH.nwafu-6A.1* (**b**) on plant height (PH). + and − represent lines with the alleles from the Icaro and Y4 of the target locus, respectively; n represent lines with the corresponding alleles, ** significant at the level of *p* < 0.01.

**Figure 5 plants-12-00847-f005:**
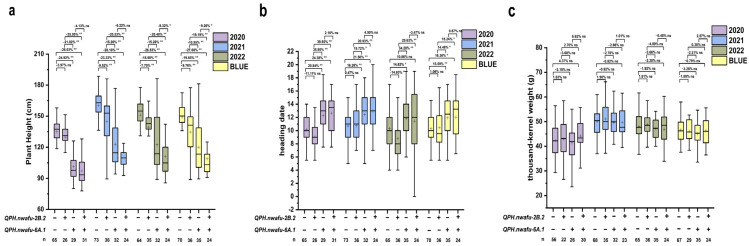
Interaction of QPH.nwafu-2B.2 and QPH.nwafu-6A.1 on plant height (PH) (**a**), heading date (HD) (**b**) and thousand-kernel weight (TKW) (**c**). + and − represent lines with the alleles from the Icaro and Y4 of the target locus, respectively; * and ** Significance at *p* < 0.05 and *p* < 0.01, respectively; ns: Not significant at *p* > 0.05.

**Figure 6 plants-12-00847-f006:**
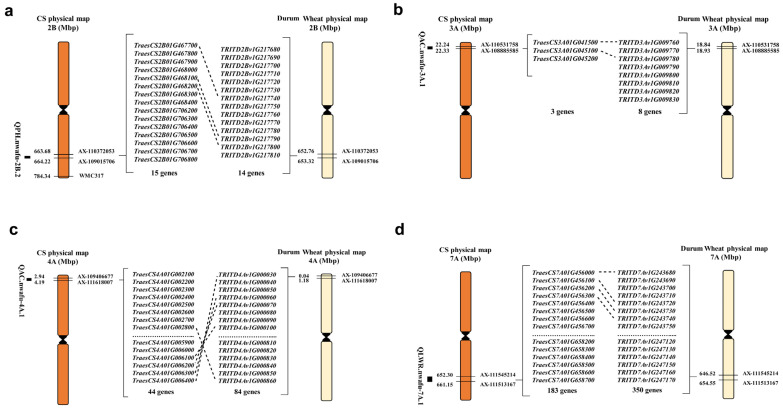
Physical maps of the *QPH.nwafu-2B.2* (**a**), *QAC.nwafu-3A.1* (**b**), *QAC.nwafu-4A.1* (**c**), and *QLWR.nwafu-7A.1* (**d**) on ‘CS’ and Durum Wheat; dotted lines indicate homologous genes.

**Table 1 plants-12-00847-t001:** Phenotypic variation and heritability of characters in different environments in parents and RILs.

Traits	Year	Parental Lines		RIL Population							
		Icaro	Y4	Min	Max	Mean	SD	CV%	Skewness	Kurtosis	*H* ^2^
**PH**	Y19	78.6	158.5	70.0	171.0	123.48	24.39	19.8	−0.425	−0.979	0.96
	Y20	80.6	156.8	77.0	160.8	119.83	22.65	18.9	−0.366	−1.182	
	Y21	81.4	168.4	81.0	188.6	143.11	26.75	18.7	−0.441	−1.091	
	Y22	83.0	159.8	85.6	186.3	136.33	24.96	18.3	−0.390	−1.100	
	BLUE	76.1	157.3	88.8	181.2	133.97	24.39	18.2	−0.433	−1.143	
**HD**	Y20	6	10	0	19	10.89	2.75	25.2	−0.241	0.713	0.83
	Y21	5	10	0	20	11.38	2.90	25.4	−0.320	0.768	
	Y22	3	8	0	19	10.48	3.38	32.2	0.022	−0.225	
	BLUE	4	9	0	18.5	10.94	2.90	26.5	−0.180	0.173	
**SL**	Y20	6.2	9.9	5.5	12.4	8.95	1.15	12.8	−0.105	0.142	0.76
	Y21	6.6	10.7	5.8	12.8	9.33	1.42	15.2	−0.053	−0.137	
	Y22	6.5	10.4	5.6	12.2	8.32	1.23	14.7	0.227	−0.081	
	BLUE	6.6	10.3	6.0	11.7	8.81	1.20	13.6	−0.114	−0.408	
**KPS**	Y20	45	60	18	64	42.25	8.27	19.6	0.179	0.391	0.50
	Y21	56	78	26	134	74.35	17.80	23.9	0.439	0.451	
	Y22	58	84	33	100	60.14	12.35	20.5	0.236	0.191	
	BLUE	56	81	36	92	58.15	9.04	15.5	0.246	0.751	
**TKW**	Y20	43.16	49.27	19.00	60.97	42.72	7.48	17.5	−0.286	0.094	0.64
	Y21	46.06	51.89	31.98	75.58	50.19	6.56	13.1	0.711	1.797	
	Y22	41.20	46.09	25.94	64.17	47.82	5.38	11.2	−0.247	1.423	
	BLUE	42.74	49.89	33.56	67.01	46.21	5.06	10.9	0.541	1.007	
**KL**	Y20	5.79	7.38	5.69	7.65	6.36	0.89	14.0	−1.344	0.931	0.41
	Y21	5.84	7.04	5.09	7.36	6.72	0.22	3.3	−0.033	0.493	
	Y22	6.40	7.57	6.19	8.21	7.28	0.42	5.8	−0.222	−0.258	
	BLUE	5.93	7.48	5.49	7.55	6.79	0.39	5.7	−0.803	0.725	
**KW**	Y20	2.69	3.37	2.63	3.43	2.99	0.39	13.1	−1.107	0.469	0.27
	Y21	2.71	3.26	2.41	3.32	2.84	0.12	4.1	0.323	3.112	
	Y22	3.10	3.44	2.83	3.96	3.41	0.20	6.0	−0.270	0.271	
	BLUE	2.87	3.41	2.65	3.40	2.98	0.16	5.5	−0.316	−0.023	
**LWR**	Y20	2.15	2.19	1.92	3.01	2.31	0.17	7.6	0.711	1.165	0.86
	Y21	2.15	2.16	1.76	2.48	2.04	0.14	6.8	0.348	−0.351	
	Y22	2.07	2.20	1.82	2.57	2.15	0.16	7.2	0.066	−0.610	
	BLUE	2.11	2.20	1.83	2.56	2.17	0.14	6.3	0.231	−0.400	
**Peri**	Y20	17.21	19.43	15.46	19.42	16.17	2.34	14.5	−1.481	1.131	0.26
	Y21	18.45	18.57	14.77	20.50	18.43	0.52	2.8	0.366	2.933	
	Y22	18.75	19.27	16.47	21.27	18.92	0.96	5.1	−0.152	−0.145	
	BLUE	18.49	19.17	15.79	20.63	18.84	0.92	4.9	−1.001	1.064	
**Area**	Y20	15.12	19.03	14.89	19.22	17.64	3.27	18.5	−1.090	0.402	0.20
	Y21	15.20	16.27	13.55	17.91	16.40	0.55	3.4	0.702	4.428	
	Y22	17.44	19.41	14.29	23.69	18.86	1.72	9.1	−0.184	0.080	
	BLUE	16.28	18.06	14.11	20.59	17.64	1.30	7.4	−0.5	0.180	
**AC**	Y19	0	2	0	2.00	0.46	0.84				0.90
	Y20	0	2	0	2.00	0.65	0.94				
	Y21	0	2	0	2.00	0.42	0.82				
	Y22	0	2	0	2.00	0.40	0.80				
**SB**	Y20	0	2	0	2.00	0.58	0.94				0.95
	Y21	0	2	0	2.00	0.72	0.96				
	Y22	0	2	0	2.00	1.45	0.90				

*SD*: standard deviation; *CV*: coefficient of variation; *H*^2^: broad-sense heritability. *PH*: plant height (cm); *HD*: heading date; *SL*: spike length (cm); *KPS*: Kernel number per spike; *TKW*: thousand-kernel weight (g); *KL*: kernel length (mm); *KW*: kernel width (mm); *LWR*: kernel length-width ratio (KL/KW); *Peri*: kernel perimeter (mm); *Area*: kernel area (mm^2^); *AC*: awn color; *SB*: spike-branching; *BLUE*: best linear unbiased estimation.

**Table 2 plants-12-00847-t002:** Correlation analysis of different traits based on BLUE.

Traits	PH	HD	SL	KPS	TKW	KL	KW	LWR	Peri	Area	AC	SB
**PH**	1											
**HD**	−0.521 **	1										
**SL**	0.004	−0.011	1									
**KPS**	0.043	0.062	0.223 **	1								
**TKW**	−0.061	0.054	−0.063	−0.558 **	1							
**KL**	−0.082	0.087	−0.257 **	−0.277 **	0.292 **	1						
**KW**	−0.148	0.155	0.025	−0.286 **	0.532 **	0.515 **	1					
**LWR**	0.041	−0.045	−0.281 **	−0.021	−0.213 **	0.556 **	−0.419 **	1				
**Peri**	−0.109	0.116	−0.234 **	−0.307 **	0.362 **	0.974 **	0.684 **	0.367 **	1			
**Area**	−0.152	0.156 *	−0.150	−0.349 **	0.529 **	0.864 **	0.854 **	0.084	0.943 **	1		
**AC**	−0.135	0.022	0.354 **	0.074	−0.095	−0.379 **	−0.076	−0.321 **	−0.346 **	−0.253 **	1	
**SB**	0.101	0.022	0.089	0.288 **	−0.219 **	−0.047	−0.100	0.046	−0.077	−0.094	−0.077	1

* Significance at the 0.05 probability level; ** Significance at the 0.01 probability level. *PH*: plant height; *HD*: heading date; *SL*: spike length; *KPS*: Kernel number per spike; *TKW*: thousand-kernel weight; *KL*: kernel length; *KW*: kernel width; *LWR*: kernel length-width ratio (KL/KW); *Peri*: kernel perimeter; *Area*: kernel area; *AC*: awn color; *SB*: spike-branching.

**Table 3 plants-12-00847-t003:** Major or stable additive QTLs in this study for the 12 traits in different environments.

Traits	QTL	Year	Interval (cM)	Left Marker	Right Marker	LOD	PVE (%)	Add
**PH**	** QPH.nwafu-6A.1 **	Y19	197.82–201.86	AX-110419766	AX-111161134	24.46	51.10	19.34
		Y20	197.82–201.86	AX-110419766	AX-111161134	47.89	61.76	20.65
		Y21	197.82–201.86	AX-110419766	AX-111161134	38.83	66.27	22.59
		Y22	197.82–201.86	AX-110419766	AX-111161134	19.73	38.24	13.08
		BLUE	197.82–201.86	AX-110419766	AX-111161134	43.49	63.87	20.88
	**QPH.nwafu-6A.2**	Y22	203.72–214.18	AX-111634639	AX-111800524	12.72	22.59	10.09
	QPH.nwafu-2B.2	Y20	165.67–166.99	AX-109015706	AX-110372053	3.84	2.40	4.09
		Y21	165.67–166.99	AX-109015706	AX-110372053	4.08	3.96	5.46
		Y22	165.67–166.99	AX-109015706	AX-110372053	4.33	6.30	5.29
		BLUE	165.67–166.99	AX-109015706	AX-110372053	6.81	5.64	6.17
**HD**	** QHD.nwafu-5A.1 **	Y21	242.25–261.17	AX-111126249	AX-110368031	10.32	14.88	1.23
		BLUE	242.25–261.17	AX-111126249	AX-110368031	7.80	16.49	1.14
	**QHD.nwafu-6A.1**	Y22	197.82–201.86	AX-110419766	AX-111161134	11.88	26.54	−1.83
	** QHD.nwafu-6A.2 **	Y21	203.72–214.18	AX-111634639	AX-111800524	8.22	11.56	−1.11
		BLUE	203.72–214.18	AX-111634639	AX-111800524	9.24	20.03	−1.29
**SL**	** QSL.nwafu-2A **	Y22	0–71.16	AX-110162878	AX-109313122	3.35	14.13	−0.51
		BLUE	0–71.16	AX-110162878	AX-109313122	3.04	10.25	−0.41
	** QSL.nwafu-3A **	Y21	223.21–244.18	AX-111762953	AX-109078462	19.53	23.02	0.78
		Y22	223.21–244.18	AX-111762953	AX-109078462	8.59	12.06	0.48
		BLUE	223.21–244.18	AX-111762953	AX-109078462	11.16	16.93	0.53
	**QSL.nwafu-4A.2**	Y21	303.85–305.14	AX-111040045	AX-110567219	16.49	20.47	−0.74
	** QSL.nwafu-4A.3 **	Y22	305.14–309.21	AX-109849268	AX-109369221	4.97	13.23	−0.50
		BLUE	305.14–309.21	AX-109849268	AX-109369221	12.19	21.25	−0.59
	QSL.nwafu-4A.4	Y22	380.01–381.92	AX-111067027	AX-109393001	4.30	5.76	0.34
		BLUE	380.01–381.92	AX-111067027	AX-109393001	5.20	4.94	0.30
	QSL.nwafu-6A	Y21	196.86–197.82	AX-108958326	AX-110419766	7.77	6.03	0.41
		BLUE	197.82–201.86	AX-110419766	AX-111161134	4.10	3.78	0.26
**KPS**	** QKPS.nwafu-2B.3 **	Y21	253.45–261.6	AX-110430429	AX-109272515	10.32	10.64	−8.18
		BLUE	253.45–261.6	AX-110430429	AX-109272515	4.16	5.51	−2.77
	**QKPS.nwafu-4B.2**	Y22	97.11–104.58	AX-109617174	AX-109533971	4.31	11.48	4.06
	**QKPS.nwafu-4B.3**	BLUE	140.19–148.79	AX-110070905	AX-111539361	6.46	12.09	−4.35
	**QKPS.nwafu-5B.3**	Y20	207.16–220.24	AX-110502417	AX-109487166	5.07	15.75	3.27
**TKW**	** QTKW.nwafu-1B.1 **	Y22	84.86–87.74	AX-109890962	AX-110926323	15.18	17.05	−2.73
		BLUE	84.86–87.74	AX-109890962	AX-110926323	7.98	8.53	−1.92
	**QTKW.nwafu-2B.3**	BLUE	376.91–390.54	AX-109281639	AX-111105832	10.27	11.45	2.27
	** QTKW.nwafu-3B.2 **	Y20	181.92–183.61	AX-108884614	AX-108894802	2.59	9.61	−2.16
		BLUE	181.92–183.61	AX-108884614	AX-108894802	10.50	12.29	−2.36
	**QTKW.nwafu-5B.2**	Y21	134.2–138.55	AX-110907796	AX-110093465	7.24	11.11	−2.34
**KL**	**QKL.nwafu-6A**	BLUE	92.81–94.79	AX-109915394	AX-108914846	9.08	16.04	−0.17
	**QKL.nwafu-7A.1**	Y21	148.15–159.54	AX-110077383	AX-110632223	10.70	18.32	−0.12
	**QKL.nwafu-7A.2**	Y22	255.14–270.77	AX-111157442	AX-110558104	5.74	13.14	−0.15
	** QKL.nwafu-5B.1 **	Y21	223.53–232.19	AX-110404958	AX-109941486	5.09	6.61	−0.07
		Y22	223.53–232.19	AX-110404958	AX-109941486	10.39	20.41	−0.17
	**QKL.nwafu-5B.2**	Y20	266.74–278	AX-109040704	AX-110503586	4.56	16.00	−0.35
	**QKL.nwafu-5B.3**	BLUE	296.97–298.76	AX-111055910	AX-111091795	10.44	14.47	−0.16
	**QKL.nwafu-6B**	Y22	275.64–280.48	AX-110667505	AX-110522948	4.43	11.76	−0.14
**KW**	**QKW.nwafu-3A**	Y22	273.22–286.42	AX-108870115	AX-108920525	3.91	11.44	0.08
	**QKW.nwafu-6A**	Y22	310.88–360.8	AX-108781069	AX-108929747	4.11	10.99	−0.08
	**QKW.nwafu-4B.2**	Y21	112.72–113.07	AX-111221393	AX-111254019	6.82	11.11	0.05
**LWR**	** QLWR.nwafu-3A.2 **	Y20	336.79–358.98	AX-111806403	AX-109925527	6.40	8.96	−0.08
		BLUE	336.79–358.98	AX-111806403	AX-109925527	8.50	11.19	−0.05
	** QLWR.nwafu-7A.1 **	Y21	166.66–175.46	AX-111545214	AX-111513167	9.10	12.12	0.06
		Y22	166.66–175.46	AX-111545214	AX-111513167	13.12	26.22	0.08
		BLUE	166.66–175.46	AX-111545214	AX-111513167	10.05	12.63	0.06
	QLWR.nwafu-1B.4	Y21	471.93–473.05	AX-111618813	AX-110394873	7.24	8.13	−0.05
		BLUE	473.05–504.63	AX-110394873	AX-111114335	6.96	7.76	−0.05
	**QLWR.nwafu-5B.5**	Y21	305.92–313.88	AX-110560995	AX-109307903	2.56	10.99	−0.05
	QLWR.nwafu-6B	Y22	155.31–157.29	AX-89438665	AX-109417511	3.16	4.85	−0.03
		BLUE	155.31–157.29	AX-89438665	AX-109417511	5.45	5.39	−0.04
**Peri**	**QPeri.nwafu-7A.1**	Y21	148.15–159.54	AX-110077383	AX-110632223	2.95	15.11	−0.17
	**QPeri.nwafu-7A.2**	Y22	255.14–270.77	AX-111157442	AX-110558104	4.22	12.57	−0.36
	**QPeri.nwafu-3B**	Y22	191.33–199.4	AX-110989965	AX-109858559	6.98	13.48	−0.35
	**QPeri.nwafu-5B.1**	Y22	220.24–222.15	AX-109487166	AX-111002136	8.59	16.15	−0.38
	**QPeri.nwafu-5B.2**	Y20	278–282.23	AX-110503586	AX-111034200	4.45	13.82	−0.87
**Area**	**QArea.nwafu-1B.1**	BLUE	52.48–62.09	AX-111221126	AX-109449137	4.36	12.90	−0.48
	**QArea.nwafu-1B.2**	Y22	74.29–79.16	AX-108788663	AX-109304925	7.33	13.51	−0.75
	**QArea.nwafu-5B.2**	Y22	207.16–220.24	AX-110502417	AX-109487166	8.45	15.40	−0.77
**AC**	QAC.nwafu-3A.1	Y21	419.89–420.52	AX-108885585	AX-110531758	8.60	8.12	0.29
		Y22	419.89–420.52	AX-108885585	AX-110531758	7.93	7.90	0.29
	QAC.nwafu-3A.2	Y21	441.43–451.72	AX-111212311	AX-110977533	6.54	6.05	−0.27
		Y22	427.82–441.43	AX-111634183	AX-111212311	5.32	5.50	−0.26
	QAC.nwafu-4A.1	Y21	424.31–447.4	AX-109406677	AX-111618007	6.11	8.92	0.32
		Y22	424.31–447.4	AX-109406677	AX-111618007	4.00	7.01	0.28
	**QAC.nwafu-4A.2**	Y20	447.4–491.27	AX-111618007	AX-109316763	4.56	13.72	0.60
	** QAC.nwafu-1B.3 **	Y19	450.74–471.93	AX-109437886	AX-111618813	54.49	5.38	0.94
		Y20	450.74–471.93	AX-109437886	AX-111618813	26.39	21.75	0.75
	** QAC.nwafu-1B.4 **	Y21	473.05–504.63	AX-110394873	AX-111114335	24.32	31.77	0.70
		Y22	473.05–504.63	AX-110394873	AX-111114335	15.10	28.50	0.69
**SB**	** QSB.nwafu-2B.1 **	Y21	253.45–261.6	AX-110430429	AX-109272515	8.11	10.00	−0.39
		Y22	253.45–261.6	AX-110430429	AX-109272515	16.46	11.20	−0.60
	** QSB.nwafu-2B.2 **	Y21	261.6–268.73	AX-109272515	AX-111680306	5.59	8.97	0.38
		Y22	261.6–268.73	AX-109272515	AX-111680306	53.74	23.16	−0.94
	**QSB.nwafu-5B**	Y21	125.85–134.2	AX-111508809	AX-110907796	3.33	15.55	−0.49

LOD: likelihood of odds; PVE (%): proportion of phenotypic variation of the corresponding QTL. Add: additive effect, Positive additive effects indicate that alleles from Y4 enhance corresponding trait values, and negative additive effects indicate that alleles from Icaro enhance corresponding trait values; The underlined QTLs indicate that they are the stable QTLs identified in this study. The bold font QTLs indicate that they are the major QTLs identified in this study.

**Table 4 plants-12-00847-t004:** Major or stable QTL mapping results of multiple environment analysis.

Traits	cQTL	Position	Interval (cM)	Left Marker	Right Marker	LOD	LOD (A)	LOD (A by E)	PVE	PVE (A)	PVE (A by E)	Add
**PH**	cQPH.nwafu-6A.2	198	197.82–201.86	AX-110419766	AX-111161134	115.63	93.28	22.35	53.57	51.32	2.25	16.12
	cQPH.nwafu-6A.3	206	203.72–214.18	AX-111634639	AX-111800524	13.77	8.35	5.42	4.27	2.61	1.67	3.66
	cQPH.nwafu-2B.2	166	165.67–166.99	AX-109015706	AX-110372053	13.31	11.17	2.13	3.83	3.55	0.28	4.25
**HD**	cQHD.nwafu-5A	261	242.25–261.17	AX-111126249	AX-110368031	10.94	7.33	3.61	10.74	8.25	2.49	0.78
	cQHD.nwafu-6A.2	200	197.82–201.86	AX-110419766	AX-111161134	11.21	7.65	3.56	15.69	8.70	6.99	−0.82
	cQHD.nwafu-6A.3	204	203.72–214.18	AX-111634639	AX-111800524	8.53	5.26	3.27	8.06	5.89	2.17	−0.68
**SL**	cQSL.nwafu-2A	69	0–71.16	AX-110162878	AX-109313122	4.19	3.97	0.22	4.70	4.54	0.16	−0.22
	cQSL.nwafu-3A	244	223.21–244.18	AX-111762953	AX-109078462	20.28	20.27	0.01	26.77	26.42	0.35	0.53
	cQSL.nwafu-4A.1	304	303.85–305.14	AX-111040045	AX-110567219	8.31	5.03	3.28	10.12	5.61	4.51	−0.24
	cQSL.nwafu-4A.2	309	305.14–309.21	AX-109849268	AX-109369221	3.36	1.81	1.56	3.82	2.15	1.67	0.15
	cQSL.nwafu-4A.3	381	380.01–381.92	AX-111067027	AX-109393001	5.90	5.54	0.36	6.89	6.59	0.31	0.27
	cQSL.nwafu-6A.2	197	196.86–197.82	AX-108958326	AX-110419766	8.09	7.37	0.72	10.31	8.60	1.71	0.30
**KPS**	cQKPS.nwafu-2B.3	261	253.45–261.6	AX-110430429	AX-109272515	10.77	8.69	2.08	21.49	10.10	11.39	−3.08
	cQKPS.nwafu-4B	103	97.11–104.58	AX-109617174	AX-109533971	5.28	3.80	1.48	6.26	4.26	2.00	1.99
	cQKPS.nwafu-5B.1	140	138.55–140.4	AX-110093465	AX-109348971	5.17	3.10	2.07	9.91	3.58	6.32	1.88
	cQKPS.nwafu-5B.2	211	207.16–220.24	AX-110502417	AX-109487166	6.23	1.38	4.85	3.47	1.33	2.14	1.16
**TKW**	cQTKW.nwafu-1B.1	85	84.86–87.74	AX-109890962	AX-110926323	5.61	3.12	2.49	6.18	4.15	2.04	−0.90
	cQTKW.nwafu-3B	183	181.92–183.61	AX-108884614	AX-108894802	3.08	2.54	0.54	5.29	3.22	2.07	−0.81
**KL**	cQKL.nwafu-6A.4	93	92.81–94.79	AX-109915394	AX-108914846	3.13	2.54	0.59	4.02	2.49	1.54	−0.07
	cQKL.nwafu-7A.3	149	148.15–159.54	AX-110077383	AX-110632223	5.33	1.09	4.25	1.35	1.20	0.15	−0.05
	cQKL.nwafu-7A.4	255	230.08–255.14	AX-109561279	AX-111157442	6.44	2.23	4.21	3.49	2.70	0.79	−0.08
	cQKL.nwafu-5B.1	224	223.53–232.19	AX-110404958	AX-109941486	13.49	2.81	10.67	5.65	3.44	2.20	−0.08
	cQKL.nwafu-6B	281	280.48–299.16	AX-110522948	AX-111218215	4.43	1.33	3.10	2.45	1.55	0.90	−0.06
**KW**	cQKW.nwafu-3A	286	273.22–286.42	AX-108870115	AX-108920525	4.64	1.87	2.77	2.94	2.14	0.80	0.03
	cQKW.nwafu-6A	312	310.88–360.8	AX-108781069	AX-108929747	4.83	1.48	3.35	3.01	1.70	1.30	−0.03
	cQKW.nwafu-4B.2	113	112.72–113.07	AX-111221393	AX-111254019	7.60	2.75	4.84	4.39	3.27	1.12	0.04
**LWR**	cQLWR.nwafu-7A.1	167	166.66–175.46	AX-111545214	AX-111513167	17.74	12.84	4.91	10.64	8.36	2.27	0.04
	cQLWR.nwafu-1B.5	473	471.93–473.05	AX-111618813	AX-110394873	8.50	4.28	4.22	5.50	2.77	2.74	−0.03
	cQLWR.nwafu-6B.1	157	155.31–157.29	AX-89438665	AX-109417511	7.70	6.33	1.37	4.50	4.00	0.50	−0.03
**Peri**	cQPeri.nwafu-7A.1	149	148.15–159.54	AX-110077383	AX-110632223	7.62	0.95	6.67	1.64	1.25	0.39	−0.11
	cQPeri.nwafu-7A.2	255	230.08–255.14	AX-109561279	AX-111157442	4.39	1.17	3.22	2.57	1.75	0.83	−0.15
	cQPeri.nwafu-3B	192	191.33–199.4	AX-110989965	AX-109858559	7.90	3.02	4.88	5.57	4.30	1.27	−0.21
	cQPeri.nwafu-5B.1	221	220.24–222.15	AX-109487166	AX-111002136	9.61	1.33	8.29	4.60	1.96	2.64	−0.14
	cQPeri.nwafu-5B.2	278	278–282.23	AX-110503586	AX-111034200	4.82	3.03	1.79	18.30	4.72	13.58	−0.21
**Area**	cQArea.nwafu-1B	77	74.29–79.16	AX-108788663	AX-109304925	7.93	2.21	5.72	5.25	2.46	2.79	−0.25
	cQArea.nwafu-5B.2	220	207.16–220.24	AX-110502417	AX-109487166	8.83	1.79	7.03	7.12	2.33	4.80	−0.24
**AC**	cQAC.nwafu-3A.1	420	419.89–420.52	AX-108885585	AX-110531758	20.51	14.29	6.21	10.04	8.20	1.84	0.21
	cQAC.nwafu-3A.2	442	441.43–451.72	AX-111212311	AX-110977533	13.68	6.91	6.76	6.28	3.88	2.39	−0.15
	cQAC.nwafu-4A.2	445	424.31–447.4	AX-109406677	AX-111618007	11.35	10.25	1.10	5.86	5.53	0.32	0.18
**SB**	cQSB.nwafu-2B.1	260	253.45–261.6	AX-110430429	AX-109272515	23.25	22.77	0.48	22.52	21.91	0.62	−0.43
	cQSB.nwafu-2B.2	268	261.6–268.73	AX-109272515	AX-111680306	11.65	11.65	0.00	12.67	12.62	0.05	0.33

LOD: likelihood of odds. cQTL: represents QTL identified via combined QTL analysis of multi-environment trials. LOD (A) and LOD (A by E): indicate the LOD value for additive and dominance effects and LOD score for additive and dominance by environment effects, respectively; PVE (A) and PVE (A by E): represent phenotypic variation explained by additive and dominance effects and additive and dominance by environment effects for corresponding QTL, respectively. The underlined cQTLs indicate that they are the stable QTLs detected using the BIP method in this study.

## Data Availability

The data presented in this study are available in this article and the Appendix A.

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
