# Peer review of "QTL Mapping for Important Agronomic Traits Using a Wheat55K SNP Array-Based Genetic Map in Tetraploid Wheat"

_plants, 2023, doi:10.3390/plants12040847_

Round 1

Reviewer 1 Report

The research built a genetic map by SNP array data from 165 recombinant inbred lines which were constructed by crossing two tetraploid wheat materials. Then QTLs for 12 agronomic traits and QTL by environment as well as epistatic effects analysis were analyzed. The aim in this manuscript was well addressed with concrete experiment and analysis, the results were also well addressed with fully developed discussion. Here, I only have a few small suggestions as following:

1.     In figure 4, please add sample number for each boxplot.

2.     During describe results for figure 6, in line 568, line 580 and line 588, please provide the specific index, eg. Figure 6a instead of figure 6, for each traits.

      3.     In figure3, the front size is too small to distinguish values on it, what is the value in the middle of edges? Legend is also too simple to explain those values. Can you modify it? Also, can you also highlight the links between stable QTLs in the plot? I notice that the identified the number of identified digenic epistatic QTLs are highly variable between traits, for example, PH and AC have over 90 pairs of epistatic relations while SL only have 3, and it seems to have no correlation with the number of QTLs detected by ICIM-BIP method, can you explain what is main reason for this difference?

      4.     In line 480-481, it says “lines carrying QPH.nwafu-6A.1 had approximately 27% shorter in height”. This expression is problematic, since QPH.nwafu-6A.1 is just the name of QTL if I understand correctly here, the specific parental genotype should be provided here. This kind of confusing expression is also seen in line 490 for “QPH.nwafu-2B.2 had a tendency to accelerate HD and reduce PH”. 

Author Response

We are so grateful for your kind question. Please see the attachment.

Reviewer 2 Report

See my suggestions given in attached pdf. Nice work

Author Response

Thank you for this valuable feedback. Please see the attachment.

Reviewer 3 Report

The manuscript “QTL mapping for important agronomic traits using a Wheat55K ANP array-based genetic map in tetraploid wheat” gives new information about QTL mapping for agronomic traits (plat height, heading date, awn color, and so on) in wheat. I think that it was required minor revision for publication.

1. tables and supplemental tables

  Please check the order of “Traits” including note.

2. Line117-118

  “TKW, KPS” → “KPS, TKW”

3. Line 117-118

  Authors describe ‘Y4 had higher PH, SL, TKW, KPS and larger kernels”. Which data show “larger kernel? Authors should add higher KL and KW.

4. Line 120

  “both parents also showed phenotypic variations in AC and SB” should be changed to concrete expression.

5. Line159-162

  LWR has significant correlation with KL and Peri.

6. 2.5 QTLmapping analysis

  Authors classify the QTL into 3 types – additive QTLs, major QTLs and stable QTLs, but they do not be weighted.

7. table 3 and Supplemental table S3

  Please make correspond the order of QTL with explanation in the text.

8. Line 241-242

  Please add the description about QSL.nwafu-4A.4 and QSL.nwafu-6A.

9. Line 248

  “from 2.23 to 10.32” → “from 2.53 to 10.32”

10. Line 253

  Please check “average of 8.50%”.

11. Line 297-300

  “Three major QTLs” → “Six major QTLs”

  Please add QAC.nwafu-1B3 as stable QTLs.

12. Line 334, 337

  Authors use “yield traits” or “yield-related traits”, but they define “spike-related traits” in line 137.

13. Line 334-335

   SB has significant correlation with only TKW.

14. Line 352

  Why aren’t many QTLs detected in plural different years, though they were less susceptible to environmental influence?  

15. Line 372-373

  Please check “explaining 0.31% to 7.31%”

16. Line 375

  Which data is  basis for “51.8%?

Author Response

(The authors gave the same response as above.)
